# From Evidence to Knowledge: A Hierarchical Probabilistic Model of the Scientific Knowledge Landscape at Web Scale

**Yaniv Slor Futterman[1]    Binyamin Perets[1]    Mark Kozdoba[1]    Shie Mannor[1,2]**
[1]Technion – Israel Institute of Technology, Haifa, Israel    [2]NVIDIA

## Abstract

Scientific literature contains essential but often fragmented and conflicting evidence, a permanent challenge brought into focus by the emergence of Large Language Models (LLMs) that can read and extract information at web-scale. Traditional methods for knowledge integration rely on knowledge graphs that treat extracted statements as deterministic facts, imposing rigid assumptions such as the closed-world assumption and independence of relationships, which fail to capture uncertainty or reconcile contradictions. We introduce a shift from deterministic fact aggregation to a probabilistic framework that models article-level evidence as noisy, partial observations of a latent hierarchical structure. Applied to a biomedical corpus, our method synthesizes article-level evidence to form stable and biologically coherent clusters, indicating that stable signals can be extracted even when inputs are sparse, biased, or unreliable.

## 1 Introduction

Scientists face a relentless challenge of integrating an ever-expanding body of evidence into a coherent understanding of complex systems. Automated web-scale LLMs, designed for knowledge extraction Garcia et al. (2024), offer a potential rescue, promising to synthesize information at a scale far beyond human capacity. However, their effectiveness is immediately confronted by a harsh reality: the scientific knowledge they are tasked with processing is inherently fragmented, noisy, and often contradictory. This is especially true in biomedicine, where an agent must make sense of thousands of articles reporting qualitative, context-specific associations with variable reliability.

To structure this information, the dominant paradigm has been to extract relational triples from text and assemble them into deterministic knowledge graphs (KGs), where nodes represent entities and edges represent relations (e.g., `Nitrate reduction decreases intestinal inflammation`, or `Visceral fat improves insulin signaling`). This approach forces the autonomous extractor (which, from now on, we refer to as "agent") to treat knowledge as a fixed set of verifiable facts and relies on two key assumptions ill-suited for scientific domains: the Closed-World Assumption (CWA) that unmentioned facts are false (i.e., have a probability of 0), and the tuple-independence assumption, which treats each relation as probabilistically independent of others. These assumptions are deeply problematic in fields where knowledge is dynamic, incomplete, and interdependent. For instance, if an agent processes evidence for (A, positively correlated, B) and (B, positively correlated, C), a deterministic KG would treat these as independent, while a more suitable model would allow the capture of a transitive structure inferring A likely influences C indirectly.

In this work, we propose a fundamental shift in knowledge representation: from deterministic graphs to probabilistic models of latent relational structure aimed at reliable inference under imperfect evidence. Rather than treat textual assertions as ground-truth facts, we model them as noisy, sparse observations drawn from an underlying, unobserved distribution over relationships. We demonstrate

39th Conference on Neural Information Processing Systems (NeurIPS 2025) Workshop: .

our approach using an agentic framework in which specialized agents mine thousands of biomedical articles to extract relational evidence (separate ongoing work, used here only for evaluation and is not part of this contribution). Inspired by the framework of Probabilistic Knowledge Bases (PKBs), we reject both the CWA and tuple-independence assumptions in favor of a joint distribution over all possible relations, where each piece of evidence serves to update a global probabilistic model in a way that remains stable under missing or biased inputs.

A natural way to model relationships between entities is through correlation. However, extracted textual evidence is neither symmetric nor statistically grounded, making it incompatible with a covariance interpretation. We therefore project the evidence into the space of positive semi-definite (PSD) matrices, ensuring consistency with transitive correlations and enabling inference across unmentioned pairs. This projection is guided by curated world knowledge, providing a high-confidence backbone for relationship existence. By anchoring noisy article-level evidence to external priors, the projection resolves underdetermination and makes disparate articles mutually informative, improving robustness to spurious or strategically misleading extractions.

Moreover, to ensure our results are statistically grounded, we adhere to the principle of maximum entropy Jaynes (1957). Accordingly, we treat extracted evidence as soft constraints and otherwise prefer the most non-committal distribution consistent with the data: a zero-mean Gaussian defined solely by its second-order structure. This approach makes our model both minimal and expressive, facilitating a faithful transformation from qualitative claims to a probabilistic structure.

We address the multi-modality of scientific discourse by unifying the covariance matrices with a Jensen–Shannon (JS) divergence divergence based Hierarchical Gaussian Mixture Model (HGMM), clustering articles by latent distributions at multiple resolutions. The hierarchy captures broad thematic groupings at higher levels and finer specializations at the leaves, offering a flexible view of scientific subfields that accommodates distributional heterogeneity and shift across contexts.

Even within coherent clusters, some articles diverge from the majority or conflict with world knowledge. To this end, we introduce a validation step that compares alternative consensus summaries of each cluster to identify systematically misaligned articles. Irreconcilable cases are placed into a dedicated out-of-consensus set, yielding more reliable clusters and providing a principled way to highlight articles whose evidence fundamentally disagrees with the broader scientific consensus, including potential artifacts of bias or erroneous extraction.

Our method proceeds in four stages: (1) We assume the existence of an *Evidence Extraction Engine*, extracting associations from biomedical texts; (2) *World-Knowledge-Guided PSD Projection*, mapping these associations into covariance matrices anchored to curated priors; (3) *Hierarchical Article Clustering*, which organizes articles into a nested mixture of latent distributions; and (4) *Validation*, improving cluster coherence and isolating out-of-consensus articles under imperfect and shifting evidence (see Fig. 1).

## 2   Related Work

Early approaches for relation extraction (RE) relied on supervised models trained over limited relation types and curated corpora. The emergence of Large Language Models (LLMs) enabled powerful few-shot RE from unstructured text Beltagy et al. (2019); Lab (2022), spurring numerous new research systems. However, most pipelines still treat extracted triples as discrete facts inserted into deterministic knowledge graphs (KGs)Downey & Etzioni (2014); Fan (2025). In biomedicine, large KGs exemplify this paradigm (e.g., the NIH Biomedical KGFund (2024), which enriches triples and scores paths for semantic consistency (e.g., inferring "A increases C" via "A increases B" and "B increases C"). Despite these advances, the reasoning remains rule-and-path based rather than generative: it relies on semantic entailment and does not posit an underlying distribution from which the literature arises, implicitly treating relations as unambiguous, static, and largely independent. Recent LLM-based systems improve extraction fidelityPeng & Zhou (2024) and entity resolution, and some weight edges by frequency or textual context Xiao (2024), but they still lack a coherent probabilistic treatment of joint uncertainty and interdependence.

These shortcomings lead to "Probabilistic" Knowledge Bases, which relax the Closed World Assumption and allow uncertainty Dong et al. (2014); Niu et al. (2012); Wang (2014). Rather than viewing extracted relations as ground truth, PKBs define a probability distribution over possible

facts. However, most PKBs assume tuple independence and require schema-specific designs, limiting their ability to capture transitive or indirect dependencies between relations. Recent work has attempted Bayesian inference from LLM outputs Cui (2025), but these models often rely on strong parametric assumptions or focus on structured domains. In contrast, our approach treats literature-derived associations as conditionally dependent evidence for a latent multivariate structure, enabling transitive inference and global uncertainty modeling.

Another related field of work is Statistical Relational Learning (SRL). SRL integrates relational logic with probabilistic graphical models, enabling reasoning over structured domains where facts are uncertain and interdependent Getoor & Taskar (2007); Richardson & Domingos (2006); Huang et al. (2012). Frameworks like Markov Logic Networks Richardson & Domingos (2006) and Probabilistic Soft Logic Huang et al. (2012) allow modeling of weighted logical constraints, but often require manual rule specification or operate on fixed graph schemas. Our method shares the SRL goal of modeling relational uncertainty, but differs by forgoing explicit logic rules in favor of learning continuous latent covariance structures. This enables scalable inference from LLM-derived evidence without requiring ontology-level constraints or hand-crafted rules.

## 3 Framework and Preliminaries

Our goal is to learn a hierarchical probabilistic model of entity relationships from a large corpus of scientific literature. Let $V = \{e_i\}_{i=1}^{M}$ be the predefined set of $M$ biological entities. We use a multi-agent LLM system capable of extracting relation types from text, where we focus on a core set of three relation types, $\mathcal{R} = \{+1, -1, 0\}$, representing *positive correlation*, *negative correlation*, and *no correlation*, respectively. It is important to note that the evidence reflects the *existence* of the relationship, assumed to be statistically validated by the related article, and not the magnitude. A detailed description of the relation types is provided in Supplementary Section A.

We index each article in our corpus of $N$ documents by $i$, where each agent processes article $i$ to extract evidence about relationships between entities $V_i \subseteq V$ mentioned therein. A fundamental assumption is that all evidence from the same article constitutes partial observations from a specific mixture of latent probability distributions, as formalized in our generative model (Section 3.1). The special case of "meta-review" papers is addressed in Supplementary Section B. The evidence from article $i$ is represented by a raw evidence matrix $R_i \in \{-1, 0, +1, Null\}^{|V_i| \times |V_i|}$, where $(R_i)_{uv}$ encodes the reported relation between entities $e_u$ and $e_v$, and *Null* marks unobserved pairs; let $T_i = \{(u, v) \mid (R_i)_{uv} \neq Null\}$ be the set of observed pairs in article $i$.

At the core of our methodology is the fact that the sparse, qualitative evidence matrices are not valid covariance structures. Our process, therefore, begins with a regularized projection of the evidence matrix $R_i$ onto the space of symmetric positive semi-definite (PSD) matrices to construct a valid $|V_i| \times |V_i|$ covariance matrix, $\Gamma_i$. This projection is formulated as an optimization problem with three regularization components (see equation 3). The resulting $\Gamma_i$ serves as a single, multi-dimensional "observation" summarizing the relational information from article $i$. We next detail these terms:

(1) *Evidence Adherence*, which ensures the resulting covariance matrix remains faithful to the observed evidence $R_i$; (2) *MaxEnt Prior*, where, following the principle of maximum entropy ( Jaynes (1957)), we select the least informative distribution consistent with the data, encouraging a well-formed covariance shape, namely a zero-mean Gaussian specified only by its second-order structure; and (3) *World Knowledge Regularization*, where aggregated knowledge from established biomedical databases (e.g., STRING-DB Szklarczyk et al. (2023)) provides high-confidence priors on the existence of relationships. This "world knowledge" is encoded as a sparse set of priors $\mathcal{S} = \{(i, j, s_{ij})\}$ with $s_{ij} \in [0, 1]$ denoting the confidence in the relationship between entities $e_i$ and $e_j$. The associated regularization loss $\mathcal{L}_{\text{world}}$ penalizes deviations from these priors, anchoring the projection and resolving underdetermination.

### 3.1 The Generative Model: Hierarchical World Model

To model the distribution of relationships across the literature, we treat each article's covariance matrix $\Gamma_i$ as a *sample* from the latent world model, which we model as a Hierarchical Gaussian Mixture Model (HGMM) with zero means. This structure allows us to discover latent clusters of articles that share similar correlation patterns (e.g., "cancer metabolism", "inflammatory response")

while modeling the corpus as a nested mixture of latent distributions, capturing both broad areas of agreement at higher levels and fine-grained specializations at the $\ell$ leaves.

Since our observations, $\Gamma_i$, are themselves covariance matrices, a standard likelihood function is not readily tractable, making direct Maximum Likelihood Estimation difficult. We therefore use an adaptation of the HGMM where the likelihood of an observation $\Gamma_i$ being generated from a cluster $k$ is based on the Jensen-Shannon (JS) divergence. This approach properly models the geometry of the space of covariance matrices, treating proximity in distributional space as the measure of likelihood:

$$p(\Gamma_i|z_i = k, \Sigma_k) \propto \exp\left(-\beta \cdot JS\left(\mathcal{N}(0, \Gamma_i) \,||\, \mathcal{N}(0, \Sigma_k[V_i])\right)\right) \tag{1}$$

where $\beta$ is a scaling parameter, and $\Sigma_k[V_i]$ is the submatrix of the cluster's global covariance matrix aligned to the $V_i$ entities in article $i$. Let $\mathcal{T}$ denote the resulting dendrogram with node set $\mathcal{N}$ and leaf set $\mathcal{L}$. We construct $\mathcal{T}$ by recursively applying a flat, *weighted* JS-GMM in a top-down manner. At node $t \in \mathcal{N}$ we fit a $K_t$-component JS-GMM to all articles using sample weights $\{w_i^{(t)}\}$ (root: $w_i^{(\text{root})} = 1/N$), obtaining soft assignments $\gamma_{i,k}^{(t)}$ and local proportions $\{\pi_{t,k}\}$. For each child $c$ of $t$, article weights inherit multiplicatively as $w_i^{(c)} = w_i^{(t)}\gamma_{i,c}^{(t)}$ and the procedure recurses on $c$ until stopping criteria are met. The hierarchy thus arises by repeatedly reusing the full corpus while concentrating probability mass along each branch via inherited weights.

Each node $t$ is summarized by a global covariance $\Sigma_t \in \mathbb{R}^{M \times M}$ defined as the weighted JS barycenter of its articles under $\{w_i^{(t)}\}$, serving as the consensus estimate at that resolution. Given local proportions $\{\pi_{t,k}\}$, the leaf prior for $\ell$ with path $\mathcal{P}(\ell)$ is:

$$\Pi_\ell = \prod_{(t \to k) \in \mathcal{P}(\ell)} \pi_{t,k}, \qquad \sum_{\ell \in \mathcal{L}} \Pi_\ell = 1. \tag{2}$$

The set $\{\Sigma_\ell\}_{\ell \in \mathcal{L}}$, article weights $\{w_i^{(\ell)}\}$, and priors $\{\Pi_\ell\}$ form the primary output of the generative model, which we use for consensus estimation, validation, and downstream inference.

Finally, while the hierarchical model encourages coherent clusters, some articles may diverge from the latent distribution or conflict with curated world knowledge. To address this, we introduce a validation stage that identifies and reassigns such articles. Articles rejected from all leaves are placed in a dedicated *out-of-consensus set*, enabling systematic identification of articles that conflict with both the article-derived consensus and the curated world knowledge. We detail the stages next: *World-Knowledge-Guided PSD Projection* (Section 4.1), *Article Clustering* (Section 4.2), *Validation and Out-of-Consensus Identification* (Section 4.3) and *Evaluation* (Section 5).

## 4 Method Overview

For evidence extraction, we assume the existence of a multi-agent system (MAS) built on large language models; the specifics of the novel MAS architecture used in our work are detailed in the Evaluations section. In this system, articles are processed independently to output a structured evidence matrix. While this per-article approach is not a strict requirement of our framework, as a set of articles could be projected to a PSD matrix collectively, we find it highly advantageous. It enables massively parallel and asynchronous relation extraction and allows each article to have its own statistical meaning, independent of the global model. Consequently, since each article typically covers only a subset of biomedical concepts, the resulting evidence matrices $\{R_i\}_{i=1}^N$ differ in dimension and sparsity, reflecting article-specific coverage.

### 4.1 World-Knowledge-Guided PSD Projection

Given $\{R_i\}_{i=1}^N$, we project each $R_i$ into the space of positive semi-definite (PSD) covariance matrices using a regularized optimization procedure. This guarantees that the output encodes a valid joint distribution over entities, respects article-level observations, and induces transitive correlations. We treat observed (non-null) entries in $R_i$ as soft constraints and impose regularization terms to: (1) enforce unit variance across all entities, (2) promote independence in unobserved pairs following the

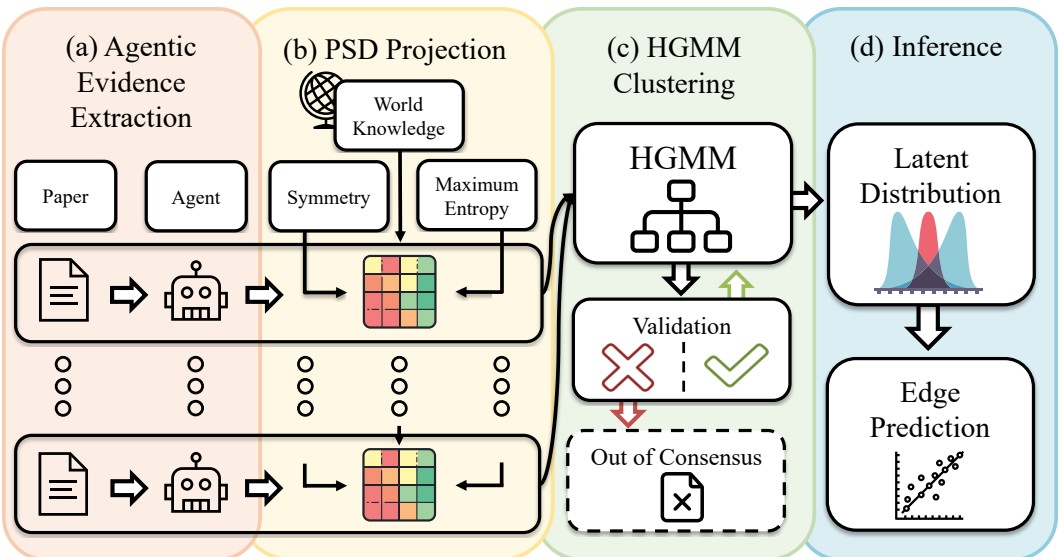

Figure 1: **Framework Overview.** **(a)** Agentic evidence extraction produces article-level relational claims structured for modeling. **(b)** World-knowledge–guided PSD projection maps claims to covariance matrices, enforcing unit variance, symmetry, and a maximum-entropy prior while aligning to curated priors from world knowledge. **(c)** JS-based hierarchical GMM clusters articles across resolutions via top-down weighted recursion, with validation that prunes misaligned articles into a dedicated out-of-consensus set. **(d)** The learned model supports downstream inference and discovery, including edge prediction for unseen entity pairs.

maximum entropy principle Jaynes (1957), and (3) encourage agreement with the curated world-knowledge matrix $S \in \left([0,1] \cup \{\text{Null}\}\right)^{M \times M}$. Let $S_i$ denote the subset of non-null entity pairs from $S$ that appear in $V_i$. Because $S$ encodes confidences in the *existence* of relationships but not their sign, we compare absolute values of covariances against $S$ (see Supplementary Section C).

We parameterize the PSD matrix as $C = A^\top A$, with $A \in \mathbb{R}^{|V_i| \times |V_i|}$, ensuring positive semi-definiteness by construction. The projected covariance $\Gamma_i$ is obtained by minimizing:

$$A = \arg \min_{C = A^\top A} \left\{ \frac{1}{|T_i|} \sum_{(u,v) \in T_i} (C_{uv} - (R_i)_{uv})^2 + \mathcal{L}_{world}(C; S) + \mathcal{R}_{struct}(C) \right\}, \quad \Gamma_i = C \quad (3)$$

$$\mathcal{L}_{world}(C; S) := \frac{\lambda_1}{|S_i|} \sum_{(u,v) \in S_i} (|C_{uv}| - S_{uv})^2 ; \mathcal{R}_{struct}(C) := \frac{\lambda_2}{|V_i|} \|\text{diag}(C) - \mathbf{1}\|^2 + \frac{\lambda_3}{|V_i|^2} \|C - I\|_F^2$$

Where $\lambda_i$ denote regularization coefficients. The main term preserves fidelity to article evidence, $\mathcal{L}_{world}$ enforces alignment with curated world-knowledge, and $\mathcal{R}_{struct}$ consists of two components: (1) a unit-variance term designed to fix $\text{diag}(C) = \mathbf{1}$ to ensure a valid standardized covariance, and (2) a maximum entropy term which promotes independence between unobserved pairs following the maximum entropy principle Jaynes (1957). This yields $\Gamma_i$, a valid covariance matrix that integrates noisy article claims with stable world knowledge, forming the foundation for downstream probabilistic modeling.

### 4.2 Hierarchical Article Clustering

Unlike standard Gaussian mixture models (GMMs) defined over fixed-length vectors, our setting requires clustering covariance matrices of varying dimensions. We use a custom JS-divergence based hierarchical GMM (HGMM) where each component $k$ at node $t$ is represented by a global positive semidefinite (PSD) covariance $\Sigma_{t,k} \in \mathbb{R}^{M \times M}$ over the full entity set of size $M$. To handle

the variable dimensions, the comparison between an article's specific covariance $\Gamma_i$ (over entity subset $V_i$) and a global centroid $\Sigma_{t,k}$ is restricted to the aligned principal submatrix, $\Sigma_{t,k}[V_i]$.

Each node $t$ maintains mixture proportions $\pi_{t,k}$ for its local components, normalized such that $\sum_k \pi_{t,k} = 1$. The responsibility of article $i$ for component $k$ at node $t$ is defined via the Jensen-Shannon (JS) divergence between Gaussians (see Supplementary Section D) where $\beta$ acts as the temperature parameter:

$$\gamma_{i,k}^{(t)} = \frac{\pi_{t,k} \exp\big(-\beta\,\mathrm{JS}(\Gamma_i \,\|\, \Sigma_{t,k}[V_i])\big)}{\sum_m \pi_{t,m} \exp\big(-\beta\,\mathrm{JS}(\Gamma_i \,\|\, \Sigma_{t,m}[V_i])\big)} \tag{4}$$

The local proportions $\{\pi_{t,k}\}$ induce a *prior* mixture over leaves. For a leaf $\ell$ with path $\mathcal{P}(\ell)$ from the root to its parent, define the leaf prior mass as the product of local proportions along the path:

$$\Pi_\ell = \prod_{(t \to k) \in \mathcal{P}(\ell)} \pi_{t,k}, \qquad \sum_{\ell \in \text{Leaves}} \Pi_\ell = 1 \tag{5}$$

Each article $i$ carries an effective weight $w_i^{(t)}$ at node $t$, representing the fraction of its mass that has reached $t$. At the root, weights are initialized uniformly as $w_i^{(\text{root})} = 1/N$ for $N$ total articles. When descending to a child component $k$, the inherited weight is updated as $w_i^{(t,k)} = w_i^{(t)} \cdot \gamma_{i,k}^{(t)}$, ensuring that the leaf-level weight $w_i^{(\ell)}$ reflects the full path of assignments from the root to the leaf $\ell$. Leaf weights are defined as $w_i^{(\ell)} = w_i^{(t_\ell)} \cdot \gamma_{i,\ell}^{(t_\ell)}$, where $t_\ell$ is the parent node of $\ell$.

Each centroid $\Sigma_{t,k}$ is updated as the weighted barycenter of its assigned articles. This ensures the problem is convex by definition as a sum of convex terms. To preserve positive semidefiniteness, we employ a factorized parameterization $\Sigma_{t,k} = A_{t,k}^\top A_{t,k}$ and solve:

$$A_{t,k} = \arg \min_{A \in \mathbb{R}^{M \times M}} \sum_i w_i^{(t)} \gamma_{i,k}^{(t)} \,\mathrm{JS}\big(\Gamma_i \| (A^\top A)[V_i]\big) \tag{6}$$

The hierarchical GMM proceeds recursively in a top-down fashion, splitting nodes until reaching stopping criteria. The end result is a partition of the corpus into leaf clusters, each leaf $\ell$ associated with a centroid covariance $\Sigma_\ell$ and per-article weights $\{w_i^{(\ell)}\}$ that quantify article contributions. Together with the leaf priors $\{\Pi_\ell\}$ defined in equation 5, these quantities parameterize the leaf-level latent mixture used for posterior responsibilities, iterative validation (Section 4.3), and unseen-edge imputation (Section 5.2). For more regarding HGMM implementation see Supplementary Section E.

### 4.3 Validation and Out-of-Consensus Identification

Once the HGMM is complete, we define two representative covariance matrices for each leaf cluster $\ell$: $\Sigma_\ell^{\text{bar}}$, $\Sigma_\ell^{\text{psd}} \in \mathbb{R}^{M \times M}$, where $\Sigma_\ell^{\text{bar}}$ denotes the JS barycenter covariance of the articles assigned to $\ell$ given by the hierarchical GMM, and $\Sigma_\ell^{\text{psd}}$ denotes the joint PSD-projected covariance obtained by optimizing over all articles in $\ell$ simultaneously, with incorporation of curated world knowledge via equation 3. The JS barycenter captures how articles collectively agree under the mixture model geometry, while the joint PSD projection enforces global consistency and world-knowledge constraints. If these two summaries diverge, it signals that the cluster contains conflicting evidence.

For each unordered pair $(u, v)$ with $u \neq v$, we quantify the discrepancy between the two global matrices as $\Delta_{uv}^{(\ell)} := (\Sigma_{\ell,uv}^{\text{bar}} - \Sigma_{\ell,uv}^{\text{psd}})^2$ and obtain the standardized per-edge discrepancy defined as $\bar{\Delta}_{uv}^{(\ell)} := \Delta_{uv}^{(\ell)} / \sigma_{uv,\ell}^2$, where $\sigma_{uv,\ell}$ denotes the weighted standard deviation of the covariance values assigned to edge $(u, v)$ by all articles in cluster $\ell$ that cover it. Edges with large $\bar{\Delta}_{uv}^{(\ell)}$ highlight loci of disagreement between the two global summaries, indicating potential out-of-consensus structure.

In parallel, for each article $i$ we define its standardized per-edge residual relative to the cluster PSD projection: $\bar{d}_{uv}^{(i)} := (\Gamma_{uv}^{(i)} - \Sigma_{\ell,uv}^{\text{psd}})^2 / \sigma_{uv,\ell}^2$ where $\Gamma^{(i)}$ is the projected covariance of article $i$. This captures how strongly article $i$ disagrees with the global cluster projection.

Each article $i$ contributes only over its covered edges $V_i$. To quantify its influence on the discrepant edges of cluster $\ell$, we define:

$$R_{i,\ell} \; := \; \frac{w_i^{(\ell)}}{|V_i|} \sum_{(u,v) \in V_i} \bar{\Delta}_{uv}^{(\ell)} \, \bar{d}_{uv}^{(i)} \tag{7}$$

This captures the average standardized discrepancy contributed by article $i$, adjusted for its hierarchical weight $w_i^{(\ell)}$ and the extent to which its covariance entries diverge from $\Sigma_\ell^{\mathrm{psd}}$. During validation, each article $i$ with $R_{i,\ell} > \tau$ is removed from $\ell$, and reallocated across the remaining leaves $m \neq \ell$ in proportion to its weights $w_i^{(u)}$, with bookkeeping to prevent reassignment back to the rejected component. After reallocation, we recompute all JS barycenters as well as update $\sigma_{uv,\ell}$. This validation step is applied iteratively over all leaves until convergence (no further moves). Articles that fail to remain in any cluster are collected into a dedicated *out-of-consensus set* $\mathcal{O}$.

# 5 Evaluations

Our evaluations are designed to test three overarching goals: first, whether the framework can recover biologically meaningful clusters that align with known pathways and disease concepts; second, whether it can reliably impute unseen associations between entities, enabling the discovery of novel relationships; and third, whether the framework improves common downstream analysis results, such as pathway enrichment analysis. Here, we present preliminary results, detail the resources already allocated to future evaluations, and outline our plans ahead.

The core of our evaluation will be conducted using two complementary data sources: 1) An article-level evidence extraction engine produced by a domain-specific agentic pipeline, which extracts relations from 3,598 full-text open-access articles, and was curated and fine-tuned using human experts. The pipeline was built and aligned to extract diet-host-microbiome entities and relations. We treat this extraction layer as fixed and independent of our modeling; 2) A corpus of 180 datasets drawn from three related biological benchmarks (Buzzao et al., 2024; Geistlinger et al., 2021; Hutter & Zenklusen, 2018). Each dataset contains raw gene-expression measurements (features) for control and disease groups, as well as a list of known biological pathways that serve as ground-truth labels associated with diseases.

This structure allows us to "create evidence" by treating each dataset as an *article* and its most significant correlations associated with the condition as "reported evidence." The similarity between articles can be assessed, as datasets sharing pathways should be grouped together by our clustering algorithm. We have already constructed this compendium and sampled correlation candidates to match the distribution of our real-world literature data.

## 5.1 Preliminary Results

**Literature Extracted Relations.** To validate the extracted relations as a source for a statistical model, we adopt a simplified flat JS-based GMM (see Section 4.2), trained on article-level evidence, without world-knowledge regularizer ($\lambda_1 = 0$). We fitted the JS-GMM for T=10 independent runs using identical hyperparameters.

Figure 2a displays the M-step objective, normalized by its initial value, across the EM iterations. The deviation between runs is presented as an error band. As the figure shows, there is a monotonic decrease after initialization. This behavior indicates a convergence to stable EM fixed points. To evaluate the stability of the clustering, we identified "consensus groups": sets of articles that were consistently assigned to the same cluster across all T independent runs. Figure 2b compares the size of the five largest consensus groups to a random allocation baseline. As the figure shows, the observed groups are significantly larger than those expected by chance, rejecting the hypothesis that these groups could be formed randomly. A single permutation test for the largest group, shown in Figure 2c, further illustrates this substantial difference.

To farther validate these internal metrics, we conducted an external, biologically meaningful evaluation. This was performed using an LLM-as-judge validation with the Claude Pro API. From each cluster, we subsampled five sets of 40 entity-relation triples in the format

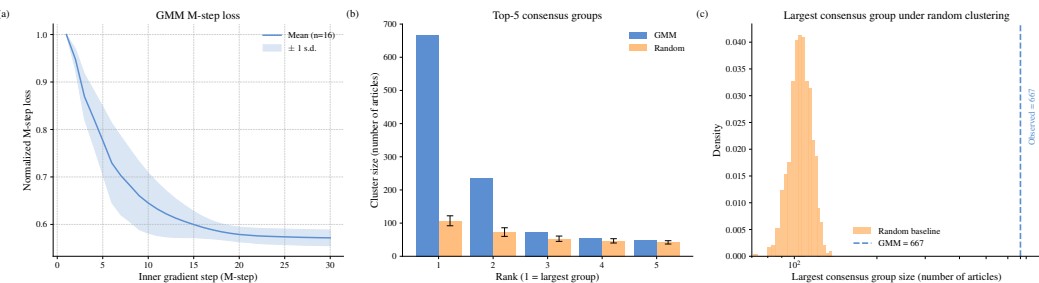

Figure 2: **Convergence and consensus across** $T=10$ **JS-GMM fits** ($K=3$). **(a)** Normalized M-step loss over inner gradient step (error-bands 5-95% in light-blue); curves show monotone descent. **(b)** *Top-5* consensus-groups from GMM (blue) versus random baseline (orange, error-bars 5-95% intervals; $R=100$ permutations); all ranks exceed the null (p<0.001). **(c)** Permutation test for *largest* consensus-group size under a random-assignment baseline ($R=1000$ permutations; log-scaled $x$-axis). Dashed line marks the observed value (667); no permutation reached this value (p<0.001).

($entity_1, entity_2, relation\_type$). The LLM was then prompted to assess the biological similarity between all pairs of these sets, assigning a score from 0 (low similarity) to 1 (high similarity). A t-test comparing the similarity scores of within-cluster pairs to those of between-cluster pairs yielded a statistically significant result (T-statistic: 35.61; $p < 0.05$). This supports the hypothesis that our clusters successfully group articles with coherent and distinct biological themes.

**Benchmark Compendium.** We next turn to the benchmark compendium with ground truth pathways. We simulate 50 articles per dataset (42 datasets; 2,100 articles; simulator details in Supplementary Section G), estimate covariances for these articles via the PSD projection, and cluster them with a flat JS-GMM using $K=10$. Figure 3 shows, for each dataset, the distribution of its articles across clusters. As the figure shows, most datasets are strongly concentrated in a single cluster.

The dashed horizontal line in Figure 3 marks the mean largest-share expected under random assignment with the same cluster-size profile, and the observed concentrations lie well above this reference. Using pathway labels as ground truth, within-cluster pathway similarity significantly exceeded between-cluster similarity ($p=4.75\times10^{-12}$), indicating that the PSD projection plus GMM recovers biologically meaningful structure at $K=10$.

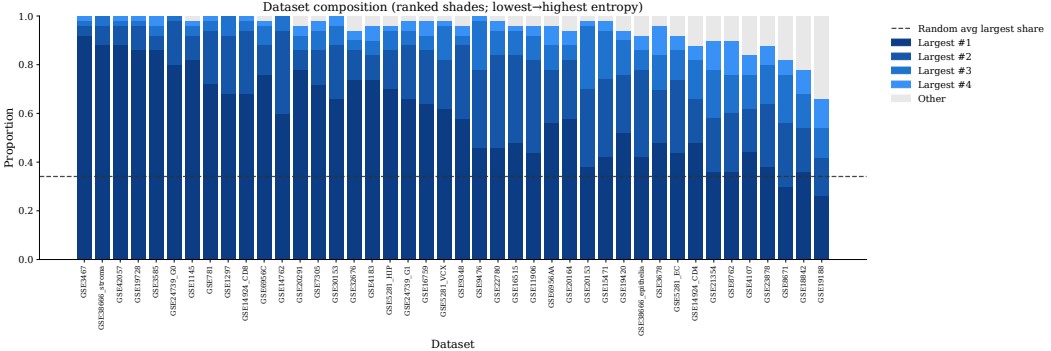

Figure 3: **Dataset composition across clusters** ($K=10$). Stacked bars show, for each benchmark dataset, the proportion of its simulated articles assigned to each cluster after PSD projection and flat JS-GMM clustering. Datasets are ordered left to right by entropy (lowest to highest). The dashed horizontal line marks the expected mean largest-share under random assignment with the same cluster-size profile. Most datasets exceed this baseline, consistent with pathway-grounded alignment ($p \leq 4.75\times10^{-12}$).

Taken together, these convergence behaviors and validation results strongly suggest that the evidence extracted by our LLM-based pipeline is internally consistent and can be faithfully represented by a statistical model.

## 5.2 Ongoing Experiments

Building on these results, we will execute a comprehensive suite of experiments.

For clustering validation, building on our preliminary pathway-grounded test using empirical covariances, we will perform a head-to-head comparison of the HGMM-derived clusters obtained from empirical covariances versus world-knowledge–guided PSD-projected matrices, evaluationg both against the ground-truth pathway in the benchmark compendium to measure our model's ability to recover biologically meaningful groupings from relational data. For the literature corpus, we will expand on the LLM-based validation with the dedicated LLM-based validation engine Wang et al. (2025).

For imputation of unseen relationships, we will employ a leave-one-out protocol on the benchmark data, training the model on all but one dataset and predicting the unseen correlations in the held-out set, and on the literature data, we will perform a similar task using two dedicated, large-scale microbiome datasets to test the model's ability to generate novel, domain-relevant hypotheses.

Finally, for the enhancement of downstream analysis, we will examine whether our framework improves standard bioinformatics workflows, especially Pathway Enrichment Analysis (PEA), a method to assess whether a set of genes is over-represented in a known biological pathway. We will compare the results of common PEA methods (e.g., GSEA Subramanian et al. (2005), Hypergeometric test) on the sparse, raw data from our benchmarks versus the same data after completion by our framework's imputation, and as an ablation study, we will determine if our framework allows for successful pathway identification from a significantly smaller initial set of known gene relationships, demonstrating an ability to amplify weak biological signals.

## 6 Conclusion

We present a framework for transforming sporadic evidence from biomedical literature into a structured probabilistic model of scientific knowledge. With the rise of agentic evidence extraction engines, meeting a long-standing necessity, we believe our world-knowledge-guided PSD projection, which turns qualitative, article-specific claims into statistically consistent covariance structures, will allow the construction of a "unified hierarchical world model" that organizes the literature at multiple resolutions. The early results are promising: we recover stable consensus clusters, and identify clusters aligned with biological knowledge. Together, they indicate that latent probabilistic models can capture reproducible signals from noisy scientific text. Moreover, we acquire an extensive set of biological benchmarks that will help with extensive validation of the usability of the framework.

### 6.1 Limitations

Our approach has three main limitations:

**Greedy hierarchy construction:** As the HGMM is learned top–down, which is computationally attractive yet introduces path dependence whereby early splits, locally optimal under current assignments, can propagate downstream, yielding leaf structures that are sensitive to initialization, stopping rules, and small perturbations in $\{\Gamma_i\}$. Our iterative validation and reassignment (Section 4.3) mitigates such greedy-bias by pruning misaligned articles, although some noise continues to persist.

**Coverage-dependent divergence:** Since cluster responsibilities and centroid updates rely on JS divergence between zero-mean Gaussians restricted to aligned subspaces, and this divergence scales with the dimensionality of the compared submatrix (i.e., with $|V_i|$), allowing higher-coverage articles to exert disproportionate influence and bias components toward such studies.

**Causal direction versus covariance modeling:** Since some of the article-level claims in $R_i$ are directional (e.g., "$A$ increases $B$"), yet within the current framework they serve as soft constraints during projection but are ultimately *interpreted as a symmetric covariance structure* in $\widehat{\Gamma}_i$, with $\mathcal{L}_{\text{world}}$ matching magnitudes to $S$ (which is direction-agnostic), an intentional abstraction that enables joint Gaussian modeling and transitive inference, but it discards causal directionality.

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

## Supplementary Material

## A  Textual Relation Taxonomy

We wish to use only those textual relations that can be *confidently* mapped to second–order statistical structure, specifically the *sign* of a covariance (positive, negative) or the explicit absence of association. We therefore define a conservative mapping from extracted relation types to $\mathcal{R} = \{+1, -1, 0\}$, where $+1$ encodes evidence for positive correlation, $-1$ for negative correlation, and $0$ for an explicit statement of no association. We distinguish $0$ from `Null`: $0$ indicates an article explicitly claims no relationship; `Null` indicates *no information* was extracted about the pair.

Let $\mathcal{A}$ be the set of textual relations returned by the agentic extractor:

$$\mathcal{A} = \{\texttt{association, NoAssociation, decreaseAssociation,}$$
$$\texttt{increaseAssociation, positiveCorrelation, negativeCorrelation,}$$
$$\texttt{causalEffect, substitution, consists}\}$$

We retain only relations that reliably indicate the *sign* of a correlation or an explicit absence claim. The mapping to $\mathcal{R}$ is:

$$\texttt{increaseAssociation, positiveCorrelation} \longrightarrow +1,$$
$$\texttt{decreaseAssociation, negativeCorrelation} \longrightarrow -1,$$
$$\texttt{NoAssociation} \longrightarrow 0$$

## B  Meta-Review Articles

Meta-reviews and broad surveys aggregate results across heterogeneous biological contexts. Treating them as if belonging to a single latent distribution is inappropriate. their evidence should act as *global prior knowledge* rather than context-specific observations. Accordingly, we treat meta-reviews as literature-derived "world knowledge" that regularizes *all* clusters.

We define the meta-review set $\mathcal{M} \subset \{1, \ldots N\}$ using document-level cues and coverage statistics. An article $i$ is flagged as a meta-review if: (1) it contains textual cues such a title/abstract containing (`review`, `systematic review`, `meta-analysis`); (2) it contains unusually broad entity coverage $|V_i|$ (e.g., above the 95th percentile of the corpus).

For each $m \in \mathcal{M}$ we compute its PSD-projected covariance $\Gamma_m$ exactly as for any article (Section 4.1), producing a coherent second-order summary of its aggregated claims.

Let $\mathcal{U}$ denote all nodes (internal and leaves) of the HGMM learned on the *non–meta-review* articles $\mathcal{A} = \{1, \ldots, N\} \setminus \mathcal{M}$. For each node $t \in \mathcal{U}$, let $\{w_i^{(t)}\}_{i \in \mathcal{A}}$ be the inherited article weights prior to meta-review injection, and let $W_u = \sum_{i \in \mathcal{A}} w_i^{(t)}$ be the total mass at $t$. We introduce a node-specific meta-review mass $\rho \in [0, 1)$ (shared across nodes) and assign to each meta-review a uniform per-node weight

$$\tilde{w}_m^{(t)} \;=\; \frac{\rho\, W_t}{|\mathcal{M}|} \quad \text{for all } m \in \mathcal{M}. \tag{8}$$

Non–meta-review weights remain unchanged, $\tilde{w}_i^{(t)} = w_i^{(t)}$ for $i \in \mathcal{A}$. This choice ensures that meta-review influence is equal *across* meta articles and that the total injected mass at node $t$ scales with its size $W_t$, while $\rho$ directly controls the strength of literature-wide regularization.

We then recompute the JS-barycenter at every node $u$ using the augmented weights:

$$\Sigma_t \;=\; \arg\min_{C \succeq 0} \sum_{i \in \mathcal{A}} \tilde{w}_i^{(t)}\, \mathrm{JS}\big(\Gamma_i \,\|\, C[V_i]\big) \;+\; \sum_{m \in \mathcal{M}} \tilde{w}_m^{(t)}\, \mathrm{JS}\big(\Gamma_m \,\|\, C[V_m]\big) \tag{9}$$

where $C[V_i]$ denotes the principal submatrix aligned to $V_i$. Operationally, this step is performed *after* the HGMM is fit on $\mathcal{A}$ and *before* the validation procedure (Section 4.3). Meta-reviews are not used to update responsibilities or to change the tree structure; they act solely as a global regularizer on node covariances.

We note that Meta-reviews are excluded from held-out folds and from discrepancy statistics in validation and edge prediction; they do not receive or trigger reassignments.

## C Coherence with world-knowledge

The curated world-knowledge evidence matrix $S$ is derived from the STRING database Szklarczyk et al. (2023), a widely used resource integrating known and predicted associations between biological entities. STRING reports both *functional* (indirect) associations such as co-expression, shared pathways, and *physical* (direct) molecular interactions supported by experimental data and curated databases. In constructing $S$, we restrict attention to the *physical interaction* subset, ensuring that the world-knowledge prior reflects experimentally grounded molecular relationships rather than indirect or correlative signals. Each selected association is assigned a confidence score in the range $s_{ij} \in [0, 1]$, reflecting the probability that the interaction exists. These scores capture existence confidence but not the magnitude or direction of the relationship. Accordingly, when incorporating $S$ into our PSD projection step (Section 4.1), we compare the absolute values of the estimated covariances to the corresponding STRING confidence scores.

## D Jensen-Shannon Divergence

During the clustering step 4.2, we use the JS divergence, a symmetric and bounded variant of KL divergence. Given two probability distributions $P$ and $Q$, the Jensen-Shannon divergence is defined as:

$$\mathrm{JS}(P\|Q) = \frac{1}{2}\mathrm{KL}(P\|M) + \frac{1}{2}\mathrm{KL}(Q\|M), \quad \text{where } M = \frac{1}{2}(P + Q) \tag{10}$$

We use the closed-form expression for KL divergence between zero-mean Gaussians:

$$\mathrm{JS}(\Gamma_i \,\|\, \Sigma_k[V_i]) = \frac{1}{2}\left[ \mathrm{tr}\left((\Sigma_k[V_i])^{-1}\Gamma_i\right) + \mathrm{tr}\left(\Gamma_i^{-1}\Sigma_k[V_i]\right) - 2|V_i| \right] \tag{11}$$

## E Hierarchical GMM

At node $t$, each article $i$ contributes a PSD-projected covariance $\Gamma_i$ on entities $V_i \subseteq [M]$, and each candidate component $k = 1, \ldots, K$ is parameterized by a global PSD covariance $\Sigma_{t,k} \in \mathbb{R}^{M \times M}$. The inherited effective weight of article $i$ at node $t$ is denoted $w_i^{(t)} \geq 0$, initialized at the root as $w_i^{(\mathrm{root})} = 1/N$ and propagated downward via $w_i^{(t,k)} = w_i^{(t)}\gamma_{i,k}^{(t)}$. We define a divergence-based pseudo–log-likelihood at node $t$ for a $K$-component mixture with weights $\{\pi_{t,k}\}_{k=1}^{K}$:

$$\mathcal{L}_{\mathrm{JS}}(K; t) \;=\; \sum_i w_i^{(t)} \log\left( \sum_{k=1}^{K} \pi_{t,k} \, \exp\left( -D(\Gamma_i \,\|\, \Sigma_{t,k}[V_i]) \right) \right) \tag{12}$$

For each internal node $t$ we evaluate $2 \leq K \leq 5$ and score a candidate $K$ using a BIC-inspired criterion,

$$\mathrm{BIC}(K; t) \;=\; -2\,\mathcal{L}_{\mathrm{JS}}(K; t) \;+\; p(K; t)\log N_u, \qquad N_t := \sum_i w_i^{(t)} \tag{13}$$

and choose

$$K^\star(t) \;=\; \arg\min_K \mathrm{BIC}(K; t) \tag{14}$$

provided the improvement over the no-split baseline is substantive, i.e.

$$\mathrm{BIC}(K^\star; t) \;\leq\; \mathrm{BIC}(1; t) - \Delta_{\mathrm{BIC}} \tag{15}$$

Each covariance $\Sigma_{t,k}$ contributes $\frac{M(M+1)}{2}$ free symmetric entries, and the mixture weights add $K - 1$ degrees of freedom, giving

$$p(K; t) \;=\; K \cdot \frac{M(M+1)}{2} \;+\; (K - 1) \tag{16}$$

The top-down recursion halts at node $t$ under three conditions: (1) if the maximum tree depth $D_{\max}$ has been reached, (2) if the node's effective mass $N_t = \sum_i w_i^{(t)}$ falls below a small threshold $\tau$, or (3) if the best split fails to improve sufficiently, that is if $\mathrm{BIC}(K^\star; t) > \mathrm{BIC}(1; t) - \Delta_{\mathrm{BIC}}$. Nodes satisfying any of these conditions are declared leaves, and their centroids $\Sigma_\ell$ together with the inherited weights $\{w_i^{(\ell)}\}$ form the final leaf-level clusters.

## F   Random Cluster Baseline

During edge prediction, in order to isolate the benefit of the act of clustering itself, we repeat each of the test folds with $R$ random clusterings matching the GMM cluster size profile.

For test fold $j$ let $\{n_k^{(j)}\}_{k=1}^K$ be the cluster sizes from HGMM on $\{d_i\}_{i \neq j}^B$. We generate $R$ random assignments of the training datasets into $K$ clusters of sizes $\{n_k^{(j)}\}$.

Each random run $r$ yields:

$$\mathcal{E}_j^{(r)} = \text{MSE over } T_{\text{novel}} \text{ using closest } \Sigma_k^{(r,j)}$$

The average random error being:

$$\mathcal{E}_{\text{random}} = \frac{1}{B} \sum_{j=1}^B \left( \frac{1}{R} \sum_{r=1}^R \mathcal{E}_j^{(r)} \right) \tag{17}$$

## G   Simulator for pathway-grounded validation

We use 42 gene-expression datasets. Each dataset $d$ has a binary *pathways vector* $\mathbf{p}_d \in \{0,1\}^P$ (ground truth). For each $d$, we run `limma` and rank genes by the moderated $t$-test $p$-value; we keep the top $k{=}80$ genes, and all sampling is from this top-$k$ set. For each dataset we generate 50 synthetic "articles": for article $r$ we sample an integer $m \sim \mathcal{N}(50, 15^2)$, sample $m$ genes without replacement from the top-$k$ pool, extract the sick-patient expression submatrix $X_d^{(r)} \in \mathbb{R}^{n_d \times m}$, and compute the empirical covariance $\Sigma_d^{(r)} \in \mathbb{R}^{m \times m}$ using the unbiased estimator with $1/(n_d{-}1)$ normalization. Across 42 datasets and 50 articles per dataset this yields 2,100 articles.

To match the modeling pipeline used for literature evidence, each empirical covariance is converted to a *binary evidence matrix* and then projected to PSD before clustering. Concretely, we form $B_d^{(r)}$ by taking the entrywise sign of $\Sigma_d^{(r)}$ off-diagonal and setting magnitude to 0.5 (i.e., $B_{uv}^{(r)} = 0.5 \, \text{sign}(\Sigma_{uv}^{(r)})$ for $u \neq v$; diagonals fixed separately), and then apply the PSD-projection objective of Eq. 3 with $\lambda_1{=}0$ (no world-knowledge regularization) to obtain $\Gamma_d^{(r)}$. We then cluster $\{\Gamma_d^{(r)}\}$ using a flat JS–GMM (Sec. 4.2).

For evaluation, each article inherits $\mathbf{p}_d$. After clustering, we draw two independent samples of article pairs, within the same cluster and between different clusters, compute the Jaccard similarity of their pathway vectors,

$$J(\mathbf{p}_i, \mathbf{p}_j) = \frac{|\mathbf{p}_i \cap \mathbf{p}_j|}{|\mathbf{p}_i \cup \mathbf{p}_j|},$$

and compare the two similarity distributions using a one-sided Mann–Whitney U (Wilcoxon rank–sum) test to assess whether within-cluster pathway similarity exceeds between-cluster similarity.

## H   Compute Resources

**Environment.** Google Colab GPU runtime with an NVIDIA A100 (40 GB VRAM); experiments executed on a single GPU worker.

**Memory footprint.** The dominant tensors are $K$ (typically 5) dense covariance-like matrices of size $20{,}000 \times 20{,}000$. Memory per matrix is $n^2 \times b$ bytes with $n{=}20{,}000$ and element size $b \in \{4, 8\}$ bytes (float32/float64). This yields $\approx 1.49$ GiB per matrix (float32) or $\approx 2.98$ GiB (float64), i.e., $\approx 7.45$ GiB (float32) or $\approx 14.9$ GiB (float64) for five matrices.

**Runtime.** PSD projection steps complete in minutes; JS-GMM clustering requires $\sim 45\text{-}60$ minutes per run. With $T{=}10$ independent fits, total wall-clock is $\sim 8\text{-}10$ GPU-hours.

**Storage.** Intermediate artifacts (matrices, logs, and figures) occupy on the order of 10-20 GB depending on numeric precision.

**Scope.** Runtimes above cover the experiments reported in the paper; small exploratory runs during development are excluded.

