# OpenReview forum: "From Evidence to Knowledge: A Hierarchical Probabilistic Model of the Scientific Knowledge Landscape at Web Scale"
_NeurIPS.cc/2025/Workshop/Reliable_ML — NeurIPS 2025 - Reliable ML Workshop_

### Official Review · Reviewer_H3va · 2025-09-10
**A promising probabilistic framework for scientific evidence aggregation with strong methodology but incomplete evaluation**

**Rating:** 7
**Confidence:** 4

**Review:**

This paper proposes a shift from deterministic knowledge graphs to probabilistic models of latent relational structure, applied to biomedical literature. The authors introduce a framework consisting of: (1) agentic evidence extraction to structure article-level claims, (2) world-knowledge-guided PSD projection that anchors noisy textual evidence to curated priors, (3) a hierarchical JS-based Gaussian mixture model (HGMM) for clustering heterogeneous covariance matrices, and (4) a validation stage that identifies and separates out-of-consensus articles. The framework aims to move beyond static fact extraction and toward robust inference under uncertainty.

Strengths
* Novelty and conceptual clarity: The paper makes a strong case against closed-world and tuple-independence assumptions, replacing them with a principled probabilistic model. This is an important and timely contribution for scientific domains characterized by incomplete and noisy evidence.
* Methodological rigor: The PSD projection step is carefully designed, combining maximum entropy principles with external world knowledge. The hierarchical GMM clustering adapts to variable-sized covariance matrices, which is technically non-trivial.
* Validation mechanisms: The proposed out-of-consensus identification adds robustness by explicitly detecting conflicting evidence, a valuable feature for practical deployments.
* Preliminary results: Experiments on biomedical corpora show meaningful clustering, stable convergence of EM, and validation against biological ground-truth, supported by error bars and permutation tests.

Weaknesses
* Evaluation is still preliminary: The results are promising but not yet comprehensive. Much of the evaluation is ongoing, and current results focus primarily on clustering stability and plausibility, with limited downstream benchmarks.
* Reproducibility gap: While datasets and methods are described in detail, the code and processed data are not yet released. This limits reproducibility for now.
* Broader impact and societal risks: The paper explicitly omits a discussion of broader impacts, which may be acceptable for foundational work, but still leaves open questions regarding potential misuse or unintended consequences in biomedical domains.
* Computational cost: Although runtimes and memory requirements are reported, the method requires significant GPU resources (∼8–10 hours per experiment). Scalability to much larger corpora remains unclear.

---

### Official Review · Reviewer_q3nH · 2025-09-15
**Peer Review of From Evidence to Knowledge: A Hierarchical Probabilistic Model of the Scientific Knowledge Landscape at Web Scale**

**Rating:** 9
**Confidence:** 4

**Review:**

Summary:
The study seeks to address some weaknesses in current LLM research related to knowledge extraction/integration from web data. The authors offer a framework to evaluate information at a probabilistic level to prevent false logical assumptions like the closed world assumption. The results imply this is a valid framework that can find well-defined concepts from noisy scientific text.

Strengths:
While the data and code used is not explicitly provided, the authors are very clear about the methods used for modeling and evaluation to be able to reproduce the experiments. Citations and description of the data (which is publicly available) is also provided. The authors state their intention to release code on acceptance. The authors also clearly state and justify their adaptation of HGMM. Figure 1 is well designed to give a high level understanding of the framework, especially to less technical audiences. The results show promising direction for this framework and sparks interest to further research.

Weaknesses:
More background could be provided on the deterministic view of the problem to offer more contrast and significance to the offered alternative. Discussion is provided on the flaws of these methods, but a few more sentences describing more detail on how these methods intend to work would help readers understand the argument for an alternative more.

Suggestions:
The paper is very well written and provides a strong argument for viewing the problem from a probabilistic perspective. My only suggestion is to add more background about how current deterministic methods work to put some context onto their stated flaws.